# Factors Influencing Long-Term Local Recurrence, Distant Metastasis, and Survival in Patients with Soft Tissue Sarcoma of the Extremities Treated with Radiotherapy

**DOI:** 10.3390/cancers16101789

**Published:** 2024-05-07

**Authors:** Arthur Lebas, Clara Le Fevre, Waisse Waissi, Isabelle Chambrelant, David Brinkert, Georges Noel

**Affiliations:** 1Radiotherapy Department, Institut de Cancérologie Strasbourg Europe (ICANS), 17 Rue Albert Calmette, BP 23025, 67033 Strasbourg, France; a.lebas@icans.eu (A.L.); c.lefevre@icans.eu (C.L.F.); i.chambrelant@icans.eu (I.C.); 2Radiotherapy Department, Léon Bérard Center, 28 Rue Laennec, 69008 Lyon, France; waisse.waissi@lyon.unicancer.fr; 3Orthopedic Surgery Department, University Hospital of Hautepierre, 1 Rue Molière, 67200 Strasbourg, France; david.brinkert@chru-strasbourg.fr; 4Faculty of Medicine, University of Strasbourg, 4 Rue Kirschleger, 67000 Strasbourg, France; 5Radiobiology Laboratory, Centre Paul Strauss, IIMIS—Imagerie Multimodale Integrative en Santé, ICube, Strasbourg University, 67081 Strasbourg, France

**Keywords:** soft tissue sarcoma of the extremities, radiotherapy, limb sparing surgery, prognostic factors, complications

## Abstract

**Simple Summary:**

This study assessed the outcomes of extremity soft-tissue sarcomas (ESTS) in 169 patients treated with radiotherapy (RT) and surgery between 2007 and 2020 in Strasbourg. Predominantly grade 2–3 tumors were found, mainly deep-seated in the lower limbs. Outcomes were promising, with 5- and 10-year LC, DC, and OS rates of 91.7%, 76.8%, and 83.8% and 84.2%, 74.1%, and 77.6%, respectively. Factors influencing the outcomes included radiation dose, grade, histologic subtype, depth and treatment technique.

**Abstract:**

Introduction: The prognostic factors for extremity soft-tissue sarcomas (ESTSs) treated with multimodal surgery and radiotherapy (RT) remain a subject of debate across diverse and heterogeneous studies. Methods: We retrospectively analyzed nonmetastatic ESTS patients treated with RT between 2007 and 2020 in Strasbourg, France. We assessed local control (LC), distant control (DC), overall survival (OS), and complications. Results: A total of 169 patients diagnosed with localized ESTS were included. The median age was 64 years (range 21–94 years). ESTS primarily occurred proximally (74.6%) and in the lower limbs (71%). Most tumors were grade 2–3 (71.1%), deep-seated (86.4%), and had R0 margins (63.9%). Most patients were treated with helical tomotherapy (79.3%). The median biologically effective dose (BED) prescribed was 75 BEDGy_4_ (range 45.0–109.9). The median follow-up was 5.5 years. The 5- and 10-year LC, DC, and OS rates were 91.7%, 76.8%, and 83.8% and 84.2%, 74.1%, and 77.6%, respectively. According to the univariate analysis, LC was worse for patients who received less than 75 BEDGy_4_ (*p* = 0.015). Deep tumors were associated with worse OS (*p* < 0.05), and grade 2–3 and undifferentiated pleomorphic sarcoma (UPS) were linked to both shorter DC and shorter OS (*p* < 0.05). IMRT was associated with longer LC than 3DRT (*p* = 0.018). Multivariate analysis revealed that patients with liposarcoma had better OS (*p* < 0.05) and that patients with distant relapse had shorter OS (*p* < 0.0001). Conclusion: RT associated with surgical resection was well tolerated and was associated with excellent long-term rates of LC, DC, and OS. Compared with 3DRT, IMRT improved local control. Liposarcoma was a favorable prognostic factor for OS. Intermediate- and high-grade tumors and deep tumors were associated with lower DC and OS.

## 1. Introduction

Approximately 1% of all adult malignancies are soft tissue sarcomas, which comprise a diverse array of tumors encompassing more than 80 histologic subtypes that grow from mesenchymal tissues [1]. The extremities are the most common site (60%), predominating in the lower limbs [1,2,3]. Standard treatment for local extremity soft tissue sarcoma (ESTS) includes at least surgery with negative margins. Radiotherapy (RT) is recommended for patients with intermediate- and high-risk ESTS [4,5].

A wide range of prognostic factors for these tumors have previously been described [6]. Positive margins, size > 5 cm, deep tumors, and high grades are the most significant factors identified in the literature that indicate a patient could benefit from RT, particularly in terms of improving LC [4,5].

However, these findings remain inconsistent across studies, and there are a limited number of studies exclusively dedicated to soft tissue sarcomas in the extremities, with even fewer prospective randomized studies available [7,8]. Moreover, the correlation between local control and survival is still disputed [6,7].

The current study was conducted to assess the long-term outcomes of radiotherapy in patients with nonmetastatic extremity soft tissue sarcomas (ESTS) at one institution and to compare our findings with those reported in the literature.

## 2. Materials and Methods

### 2.1. Study Population

We conducted a monocentric retrospective study including patients with ESTS treated with RT. A total of 169 patients were included from 2007 to 2020. The selection criteria were as follows: i. age ≥ 18 years, ii. histologic diagnosis of soft tissue sarcoma according to the *Fédération Nationale des Centres de Lutte Contre le Cancer* (FNCLCC), iii. pre- or postoperative external beam radiotherapy (EBRT), and iv. surgical resection. The exclusion criteria were i. metastatic disease; ii. nodal involvement; iii. tumor located outside the extremities, defined as from the shoulders to the hands and from the hips to the feet; iv. patients with osteosarcoma, chondrosarcoma or Ewing family tumors; and v. RT given with palliative intent.

Treatment decisions were made during multidisciplinary meetings. Histological diagnosis and grading were conducted or reviewed by a specialist pathologist at a reference center. A superficial tumor was defined as a tumor localized above the superficial fascia, while a deep tumor indicated involvement of the fascia or location beneath it.

Irradiation of grade 1 sarcoma was considered if one of the following factors was present: a tumor larger than 5 cm, R1 status removal, or nonmonobloc surgery.

### 2.2. Treatment and Irradiation Technique

Treatment modalities included 2D-RT, 3D-RT, or IMRT using helical tomotherapy. Normofractionated, hypofractionated or hyperfractionated regimens were used. Both preoperative and postoperative RT were used. Because of these different fractionations and to allow comparisons among the different schedules, the doses were converted to BEDGy_4_ and BEDGy_10_.

The gross tumor volume (GTV) was delineated on preoperative T1-weighted gadolinium-enhanced magnetic resonance imaging (RMI) in the same way as that used for pre- and postoperative RT and then realigned via CT simulation. The clinical target volume (CTV) ranged from 5 mm in all directions to 5 cm superiorly and inferiorly and 1.5 cm circumferentially; it was then corrected at the anatomical barriers, such as bone or the main fascia. For postoperative RT, the CTV included the entire surgical scar and drain orifices, as well as postoperative hematomas or collections. In most cases, a planning target volume (PTV) margin of 5 mm was added to the CTV.

Chemotherapies could be administered in various settings, including neoadjuvant, concurrent, adjuvant with radiotherapy, or at the metastatic stage, following discussion in a multidisciplinary medical meeting with a medical oncologist.

### 2.3. Follow-Up

Clinical and imaging outcomes were evaluated, mainly via MRI of the local primitive site and thoraco-abdomino-pelvic CT for metastatic screening. Clinical examination was performed by a radiation oncologist, surgeon, and medical oncologist, if necessary, every 4 to 6 months for the first 5 years, followed by annual evaluations.

### 2.4. Clinical Outcomes

Clinical outcomes included local control (LC), distant control (DC), overall survival (OS), and adverse events (AEs). Local and distant control were defined as the time intervals from diagnosis to local or distant relapse, to the last follow-up, or death, which were calculated from the first operative date, the first day of RT, or the first cycle of neoadjuvant CT, respectively, if delivered. OS was defined as the percentage of patients who were still alive at 5/10 years after diagnosis. AEs were defined as post operative complications (wound complications such as hematoma, infection, and necrosis) and lymphoedema; acute radiation-induced side effects (radiodermatitis and edema); and chronic radiation-induced side effects (fibrosis, telangiectasias and fractures). AEs were graded according to the National Cancer Institute Common Terminology Criteria for Adverse Events (CTCAE) version 5.0.

### 2.5. Statistical Analyses

LC, DC, and OS were estimated using the Kaplan–Meier method. The log-rank nonparametric test was used for comparison of survival distributions in subgroups. The alpha risk was set to 5.0%. Prognostic factors with a *p* value < 0.1 in univariate analysis and with no missing values exceeding 10% were entered into a Cox regression model for multivariate analysis. The data were checked for multicollinearity with the Belsley–Kuh–Welsch technique, and proportional hazards were checked according to the Schoenfeld residuals. No multivariate analysis could be performed for LC due to the low number of local recurrences. A two-sided *p* value < 0.05 was considered to indicate statistical significance. Statistical analysis was performed with EasyMedStat (version 3.30.2; www.easymedstat.com, France). The Institutional Review Board approved this retrospective study (IRB-2023-12). The database followed the rules of the French CNIL *(Commission National de l’Informatique et des Libertés)* MR004 model.

## 3. Results

### 3.1. Patient Population

Our institutional series included 169 patients enrolled between 2007 and 2020. The median follow-up for the entire cohort was 5.5 years. The median age was 64 years (range 21–94). The ESTSs were predominantly deep-seated (87%), located in the lower arm (71.1% %), and proximal (74.6%). The details are provided in Table 1. 

A total of 168 patients (99.4%) underwent limb-sparing surgery, while two patients underwent amputation due to tumor extension. All patients received radiotherapy, with the majority (91.7%) receiving postoperative treatment and a smaller proportion (8.3%) receiving preoperative treatment. The median time from the first symptoms to the first consultation was 4 months. The median time to initiation of postoperative RT was 82 days, while in the preoperative setting, the median time between radiotherapy and surgery was 83 days. The median doses administered for both preoperative and postoperative RT were 75 BEDGy_4_ and 60 BEDGy_10_, with doses higher than 85 BEDGy_4_ and 65 BEDGy_10_ administered for 42.6% of patients. Normofractionation was the predominant treatment regimen and was administered to most patients (91.1%). Nonstandard regimens were used in smaller proportions, with hypofractionation (daily dose of 3 Gy) for 7.7% of patients and bid-hyperfractionation (1.15 Gy twice daily) for 1.2%. The median overall treatment time was 37 days (range 20–67). 

The median clinical target volume (CTV) and planning target volume (PTV) were 446.6 cm^3^ and 699.0 cm^3^, respectively. In the 109 (64.4%) patients with available data, the median gross tumor volume (GTV) was 213 cm^3^.

Following preoperative RT, the median percentage of residual tumor cells was 50% (Q1: 0.18; Q3: 0.83), with two patients achieving a complete pathological response (0% residual tumor cells), independent of the pathological subtype. 

Chemotherapy was administered to 43 patients (25.4%), mainly based on anthracycline and ifosfamide. Of these, 16 patients (37.2%) received neoadjuvant CT, 19 patients (44.2%) received adjuvant CT prior to RT, 4 patients (9.3%) received concomitant CT, and 4 patients (9.3%) received adjuvant CT after RT. 

Patients with positive margins exhibited a significantly greater prevalence of liposarcomas (55.0% vs. 20.8%; *p* < 0.001) and low-grade tumors (52.7% vs. 14.9%; *p* < 0.01) than did those with negative margins (R0). Patients with liposarcoma had significantly greater proportions of grade 1 tumors (70.4% vs. 29.6%, *p* < 0.01).

### 3.2. Local Control

At the time of evaluation, 16 patients (9.5%) experienced local recurrence. Among them, seven (43.8%) also developed distant metastases, and eight (50.0%) died by the last follow-up. The median duration until local progression was 26.3 months (range 1.3–96.2). Local recurrence was diagnosed through imaging in 52.4% of patients and clinically in 47.6%. The 5- and 10-year LC rates were 91.7% and 84.2%, respectively. The preoperative and postoperative 5-year RT LC rates were similar at 91.7% and 91.2%, respectively (*p* = 0.391).

According to the univariate analysis (Table 2), the 5-year LC rate was significantly worse for patients who received less than 75 BEDGy_4_ (60.0% versus 92.6%; *p* = 0.015) or 60 BEDGy_10_ (71.4% vs. 93.2%; *p* = 0.046). The five-year LC was significantly greater for the IMRT technique than for the RT3D technique (94.9% vs. 78.2%, respectively; *p* = 0.018). A trend toward significance revealed higher local control (LC) rates in patients with a duration of symptoms lasting less than four months from the onset of the disease to the initial consultation (97.7% vs. 84.3%; *p* = 0.061). The five-year LC rate was greater for liposarcoma than for the other pathology subtypes but did not reach significance (98.0% vs. 88.2%; *p* = 0.08). Among the liposarcoma subgroups, only patients with dedifferentiated liposarcomas experienced local relapse, with a 5-year LC rate of 75.0%.

No difference in LC rate was observed according to sex, location, depth, grade, margin removal status, size, or use of preoperative MRI for RT treatment planning.

Local recurrences were managed with various strategies, either individually or in combination. Ten patients (66.7%) underwent a second surgery, among whom four patients underwent amputation. Reirradiation was performed in four patients (25.0%), with a median reirradiation dose of 79.3 BEDGy_4_ (range 53.1–96.1) and 66.3 BEDGy_10_ (range 31.5–74.1). Additionally, chemotherapy, mainly anthracycline and ifosfamide, was administered to seven patients (43.8%). Twelve, four, and no patients received one, two or three of these modalities, respectively.

### 3.3. Distant Control

At the last follow-up, 40 patients (23.7%) developed one or more metastases. The 5- and 10-year DC rates were 76.8% and 74.4%, respectively. The median time to metastasis was 11.6 months (range 1.1–122).

The five-year DC rate of Grade 1 tumors was 95.4%, while that of Grade 2–3 tumors was 66.6% (*p* = 0.003). The five-year DC rate was 92.9% for patients with liposarcoma and 68.1% for patients with other pathologies (*p* = 0.001). Among the liposarcoma subgroups, patients with pleomorphic liposarcoma had the worst 5-year DC rate of 33.3% (*p* < 0.001). Five-year DC was significantly worse for undifferentiated pleomorphic sarcoma (UPS) than for the other pathologies (54.8% vs. 81.1%; *p* = 0.003). The five-year DC rates for patients with and without MDM2 amplification were 95.2% and 68.1%, respectively (*p* = 0.005). Patients treated for ESTS with a dose greater than 75 BEDGy_4_ or 60 BEDGy_10_ exhibited significantly fewer metastases than those who received a lower dose, with 5-year DC rates of 77.9% vs. 40.0%, *p* = 0.037, and 78.3% vs. 42.9%, *p* = 0.022, respectively.

Patients who underwent incomplete surgery had less frequent metastases than did those with negative margins (*p* = 0.014). Compared with deep tumors, superficial tumors tended to have a better 5-year rate of DC (95.2% vs. 73.9%, *p* = 0.07). According to multivariate analysis, no prognostic factors for DC were identified (Table 3).

Among the 58 metastases, most were located in the lung (82.5%), followed by the liver, bones, and lymph nodes (Table 4). Metastases were treated locally with radiotherapy (34.2%), surgery (23.7%) or interventional radiology (10.5%). Twenty-two patients (57.9%) received metastatic chemotherapy based on anthracycline and ifosfamide or trabectedin. Front-line supportive care was initiated for four patients.

### 3.4. Overall Survival

At the median follow-up, the OS rate was 85.1% (CI: 78.8–89.7%). The 5- and 10-year OS rates were 83.8% and 77.6%, respectively.

The five-year OS rates of patients with deep tumors and those with superficial tumors were 81.2% and 100%, respectively (*p* = 0.03). As expected, five-year OS rates were worse for patients with high- and intermediate-grade tumors than for patients with low-grade tumors: 86.5% vs. 64.3%; *p* = 0.024 (Figure 1). 

The five-year OS rate was 96.6% for patients with liposarcoma compared to 77.5% for patients with other pathologies (*p* = 0.008) (Figure 2), while patients with UPS had a 5-year OS rate of 70.8%, compared to 86.7% for patients with other subtypes (*p* = 0.031). 

Among the liposarcoma subgroups, patients with pleomorphic liposarcoma had a worse 5-year OS rate (66.7%, *p* = 0.036). The five-year OS was worse for patients who developed metastasis after treatment (43.8% versus 96.1%; *p* < 0.001). Although better survival rates were found for patients with amplified MDM2, the difference was not significant (98.9% vs. 79.5%; *p* = 0.07).

Overall survival did not significantly differ according to sex, age, tumor size, tumor location, margin status, radiation dose, use of preoperative magnetic resonance imaging (MRI) or occurrence of local relapse.

According to the multivariate analysis (Table 4), the liposarcoma subtype emerged as a positive prognostic factor for survival, with a risk reduction of approximately 60% compared to that of other pathologies (HR 0.372 [0.14; 0.988], *p* < 0.05). However, metastatic development was identified as a negative prognostic factor for overall survival (HR 3.94 [2.21; 7.01], *p* < 0.0001).

### 3.5. Complications

Among the 169 patients, 48 (28.4%) experienced surgery-related complications such as hematoma, infection, necrosis, and lymphoedema. The acute radiation-induced side effects reported in the current study were radiodermatitis and edema and were observed in 136 patients (81.0%). Radiodermatitis was the most prevalent acute side effect, occurring in 131 patients (78.0%), with grade ≥3 observed in 12 patients (7.1%). Chronic radiation-induced side effects, such as fibrosis, telangiectasias and fractures, occurred in 75 patients (44.6%).

Among the acute and chronic adverse events related to RT, grade 3 adverse events, including radiodermatitis, fibrosis and telangiectasias, occurred 14 times (10.7%), one grade 4 adverse event (0.6%) corresponded to radiodermatitis, and no grade 5 adverse event was reported. Fractures were observed in five patients, with two patients having developed fractures in the femur, with a median time of appearance of 4.9 years (range 0.9–9.7) after initial diagnosis.

## 4. Discussion

The established treatment protocol for ESTS involves limb-sparing surgery, with the addition of pre- or postoperative RT for well-selected tumors with aggressive characteristics. Previous studies have demonstrated the equivalence of this approach to amputation in terms of local control, distant outcomes, and overall survival, with the added benefit of enhancing quality of life [7,9,10]. However, the literature focusing on STS of the extremities is limited, and most of these studies have a follow-up period of less than 5 years, which may not be sufficient to comprehensively assess all prognostic factors in this highly heterogeneous disease. In this study, we present a retrospective analysis of 169 patients with localized ESTS who underwent surgical treatment with RT, with a median follow-up duration of 5.5 years. Most of the tumors presented intermediate- and high-risk features, characterized by intermediate- and high-grade (G2: 27.7%; G3: 44.0%), deep location (75.5%), and size larger than 5 cm (70.4%).

At 5 and 10 years, the LC, DC, and OS rates were 91.7% and 84.2%, 76.8% and 74.1%, and 83.8% and 77.6%, respectively. The current results can be compared positively to the previous results published in the literature [6]. Indeed, previous studies including both preoperative and postoperative RT for nonmetastatic patients reported a 5-year LC ranging from 67.6% to 92.4%, a 5-year DC ranging from 42% to 87% and a 5-year OS ranging from 56% to 96% [11,12,13,14,15,16,17,18].

The median time to local relapse was 26 months, and at least 75% (Q3) of recurrences occurred before 53 months. Relapses were diagnosed through both imaging and clinical assessments in comparable proportions. These findings emphasize the significance of maintaining close radio-clinical surveillance for 5 years, as recommended in the proposed guidelines [19,20], and subsequently spacing it out, with imaging of the primary site indicated if there is clinical suspicion.

Dogan et al. reported significantly worse LC rates for patients receiving less than 60 Gy (*p* = 0.03), without impacting overall survival (OS) [21]. In the present study, we observed reduced LC rates for doses <75 BEDGy_4_ (60.0% vs. 92.6%; *p* = 0.015) and <60 BEDGy_10_ (71.4% vs. 93.2%; *p* = 0.047), with no effect on OS either, although the subgroups receiving less than 75 BEDGy_4_ and 60 BEDGy_10_ were too small to draw definitive conclusions. Higher doses (≥75 BEDGy_4_ and ≥65 BEDGy_10_) did not enhance LC, possibly because 75 BEDGy_4_ and 60 BEDGy_10_ proved sufficient for patients with R0 margins, which were predominant in our study (63.9%), with only three patients having R2 margins. Despite numerous studies demonstrating that margin status is a crucial prognostic factor for LC [16,17,22,23,24], the current study did not replicate these findings. Nevertheless, patients with incomplete margins received at least 64 Gy (96 BEDGy_4_ and 76.8 BEDGy_10_), a dose associated with improved outcomes in prior studies for this specific subgroup [25,26]. In the present study, a higher dose may mitigate the adverse impact of positive margins.

In the current study, IMRT clearly and significantly differed from 3DRT in terms of local control rates, with an 80% improvement in local failure rates at 5 years (4.3% vs. 21.8%). To our knowledge, this study is the second to demonstrate such results in ESTS, following Folkert et al., who reported an approximately 50% improvement in local failure rates at 5 years compared to 3DRT (7.6% vs. 15.1%) in a series of 319 patients with a median follow-up of 5 years. The authors explained that IMRT enables more uniform dose delivery with homogeneous coverage in the target volume, particularly for large tumors, facilitating the treatment of all tumor cells at an adequate dose [15]. Given these results and even in the absence of randomized evidence comparing IMRT with 3DRT, IMRT appears to be the preferred technique [27]. These results may also be in part thanks to simultaneous improvements in surgical techniques over time.

The impact of LC on OS remains debated across studies [22,28,29,30,31]. Local relapses can be managed through various treatments, including salvage treatments such as amputation [32], and do not necessarily lead to death. Additionally, they do not uniformly result in the development of metastases. Potential bias could stem from the very proximal locations of relapses, which may infiltrate abdominal and thoracic organs, or complete resection may be more challenging due to the emergence of neurovascular bundles. However, tumor location and depth did not influence the incidence of LC in the present study.

Since magnetic resonance imaging (MRI) is the primary imaging modality for the extremities [19], the utilization of preoperative MRI for postoperative target volume delineation has not shown significant benefits in terms of local control; this may be ascribed to the margins of the clinical target volume (CTV) taken around the surgical bed and the use of surgical clips. However, preoperative MRI is still recommended before managing these diseases, potentially allowing for the irradiation of less healthy tissue [19,33].

Concerning distant relapse, the median time to metastasis was 11.6 months, which is lower than the times reported in various other series ranging from 14 to 36 months [11,16,29,30,34,35]. A possible explanation for this difference could be the long delay between the first symptoms and the first consultation and management. Indeed, the median time before seeking medical care was 4 months. While comparable information was not available in other series, such a delay seems notably extended and represents a missed opportunity for timely intervention for the patients. This shorter delay may also be attributed to regular surveillance through thoraco-abdominal-pelvic CT scans every 4 to 6 months within our institution, facilitating the prompt detection of potential evolving lesions. The lung was the most frequent site of metastasis (82.5%), consistent with findings in the literature, reaching 84% [29]. Subsequently, but in smaller proportions (<20%), the most commonly affected organs were the bones and the liver, aligning with reports from other series [21,34,35].

We observed that most patients developed metastases within the first 5 years (5- and 10-year DC: 76.8% and 74.4%, respectively). Therefore, surveillance for distant relapses should primarily focus on the lungs through thoracic CT scans, especially in a close monitoring fashion during the initial 5 years. Recent studies have reported the feasibility of low-dose or ultralow-dose CT of chest tissue with deep learning for the detection of secondary pulmonary lesions, with encouraging results, especially for patients undergoing frequent examinations [36,37].

In the present series, patients with positive margins demonstrated a significantly lower incidence of metastases than did those with negative margins (R0). This observation could be explained by the notably greater proportion of liposarcomas and low-grade tumors in patients with positive margins, factors previously associated with lower rates of distant recurrence in this study and corroborated by other authors [11,16,30].

Other favorable results were also observed for OS. Age > 55/60 years has been identified as a significant negative factor for survival [17,29]. Although we did not observe the same result, the median age in the current study, at 65 years, was greater than that reported in other comparable series [11,30,38,39].

Low grade, which was found to be a significant positive prognostic factor for DC and OS in the current study, is known to be one of the most crucial prognostic factors for metastasis development and, consequently, a decrease in overall survival [16,17,22,30,40,41]. However, some authors have identified grade as a prognostic factor for local control [22,41]. Therefore, high tumor grades may also increase the risk of local recurrence, but this effect might be slower to develop than metastases or death, potentially explaining these discordant results.

Our study included approximately 75% of intermediate- and high-grade sarcomas, with overall survival rates that were equivalent or superior to those of series with comparable proportions of low- and high-grade sarcomas [30,38,40,42].

Deep tumors were identified as a negative prognostic factor for overall survival (OS) in univariate analysis. In a series by Goertz et al., who analyzed the UPS of the extremities, they similarly observed a significant negative impact of deep tumors on OS, although deep tumors did not affect local control (LC) [17].

Histology did not have an impact on local control (LC), consistent with findings in other studies [14,17,20]. Although not statistically significant, liposarcoma still demonstrated excellent local control of 98% at 5 years in our study, which is similar to other studies that reported a 5-year LC rate in patients with liposarcoma ≥ 96% [43,44,45]. Patients with liposarcomas had the best OS and DC among the different pathologies, and this difference persisted in multivariate analysis, with a 60% reduction in the risk of death compared to that of patients with other pathologies (HR 0.372 [0.14; 0.99], *p* = 0.047). The radiosensitivity of liposarcomas has already been demonstrated to be greater than that of other sarcoma subgroups, leading to improved oncological outcomes [43,45,46]. In this context, some authors have conducted dose reduction studies with encouraging results [47]. Pleomorphic liposarcoma, accounting for approximately 5–10% of liposarcoma subtypes, is recognized as the most aggressive subtype [48]. In the present study, 5.4% of the patients had pleomorphic liposarcomas, and they had the lowest 5-year DC and OS rates of 33.3% and 66.7%, respectively. Conversely, well-differentiated liposarcoma, the predominant subtype representing 50.0% of liposarcomas in this study, has a better prognosis and lacks metastatic potential [49]. These findings are consistent with our results showing 100% local control, distant control, and overall survival for this subgroup.

While the addition of systemic treatment may seem to be a viable option to control the disease, patients who received neoadjuvant, concomitant, or adjuvant chemotherapy with RT did not show improved oncological outcomes; this could be attributed to selection bias, as chemotherapy might have been administered to patients who were diagnosed with more aggressive disease.

The incidence of complications remained relatively low, with very few grade 3 or higher acute or chronic adverse events, as reported in other series [15,21,29,31,35,39,50]. These events, whether related to RT or surgery, remained manageable, resulting in favorable functional outcomes. Therefore, these results support the use of RT.

Limitations of this study include its retrospective, single-center design. The rarity of the disease and the limited number of patients treated over extended periods could compromise the statistical power of the study and introduce bias. Additionally, a multivariate analysis for local recurrence could not be conducted due to the limited number of patients in this subgroup.

However, our study stands out as one of the largest among the few studies presenting long-term outcomes and prognostic factors for ESTS, contributing to the advancement of knowledge about this rare and heterogeneous disease. Large prospective studies are needed to optimize personalized treatment strategies considering histological subtypes.

## 5. Conclusions

In this retrospective homogeneous study of patients with extremity soft tissue sarcoma, the combined approach of radiotherapy and surgery demonstrated good tolerability and resulted in positive long-term outcomes, including favorable rates of local and distant control, as well as overall survival. Compared with 3DRT, IMRT improved local control. Liposarcoma has emerged as a favorable prognostic factor for OS. Patients with intermediate- and high-grade tumors, as well as deep-seated lesions, exhibited significantly lower rates of distant control and overall survival.

## Figures and Tables

**Figure 1 cancers-16-01789-f001:**
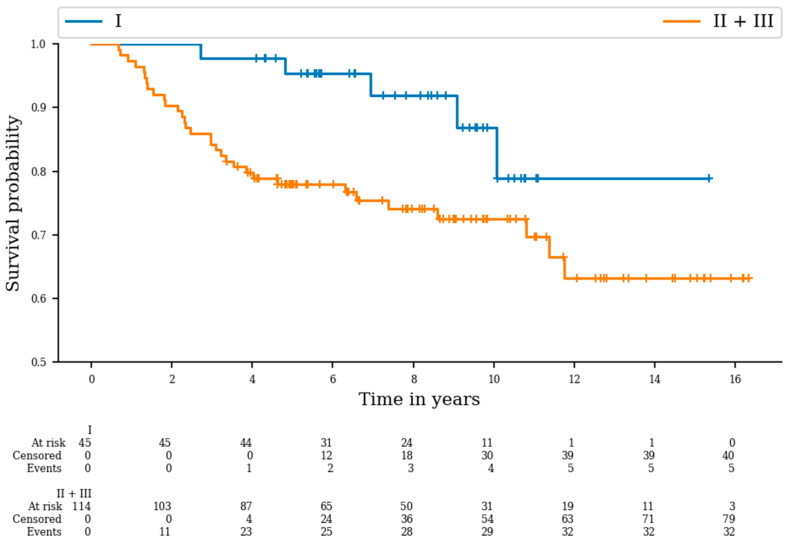
Overall survival curves according to grade (*p* = 0.024).

**Figure 2 cancers-16-01789-f002:**
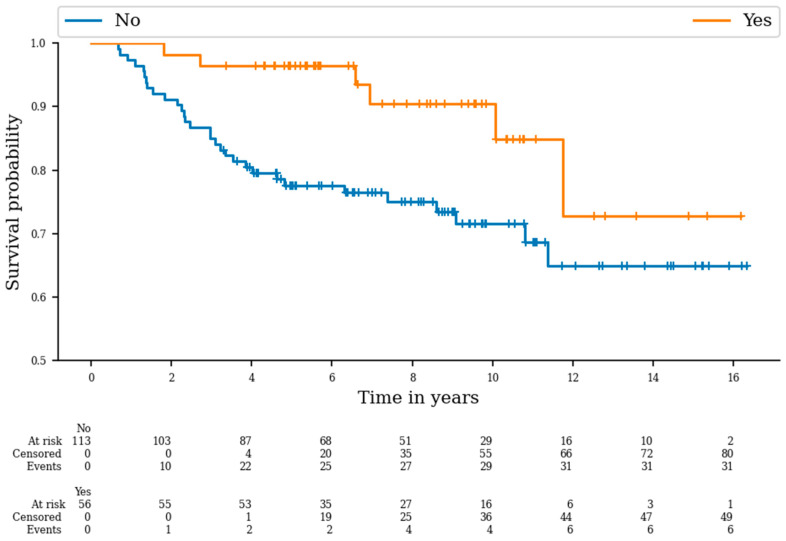
Overall survival curves of patients with and without liposarcoma (*p* = 0.008).

**Table 1 cancers-16-01789-t001:** Patient and tumor characteristics.

	Patients (n)	%
**Sex**		
Male	86	50.9
Female	83	49.1
**Age** (median)	64 (21–94)	
>60	100	59.2
<60	69	40.8
**Location**		
Upper arm	52	30.8
Lower arm	117	69.2
Proximal	126	74.6
Distal	43	25.4
**Depth**		
Superficial	23	13.6
Deep	146	86.4
**Grade (FNCLCC)**		
1	45	26.6
2	44	26.0
3	70	41.4
Unknown	10	5.9
**Margin status**		
R0	106	62.7
R1	57	33.7
R2	3	1.8
Unknown	3	1.8
**Size**		
Median size (cm)	8.0 (0.4–60.0)	
<10	101	59.8
≥10	68	40.2
**TNM**		
T1a	15	8.9
T1b	25	14.8
T2a	12	7.1
T2b	117	69.2
**Histological subtype**		
Liposarcoma	56	33.1
UPS	31	18.3
Myxofibrosarcoma	32	18.9
Leiomyosarcoma	17	10.0
Synovial sarcoma	6	3.6
Undifferentiated sarcoma	9	5.3
Rhabdomyosarcoma	4	2.4
Other	14	8.3
**Liposarcoma subgroup**		
Well-differentiated	28	50.0
Myxoid	15	26.8
Pleomorphic	3	5.4
Dedifferentiated	6	10.7
Unknown	4	7.1
**MDM2 amplification**		
MDM2+	42	24.9
MDM2−	80	47.3
Unknown	47	28.8
**RT Schedule**		
Preoperative	14	8.3
Postoperative	155	91.7
**RT Technique**		
2D-RT	1	0.6
3D-RT	34	20.1
IMRT	134	79.3
**BEDGy_4_**		
BEDGy_4_ (median)	75.0 (45.0–109.9)	
BED_4_ ≥ 75 Gy	163	96.4
BED_4_ < 75 Gy	6	3.6
BED_4_ ≥ 85 Gy	72	42.6
BED_4_ < 85 Gy	97	57.4
BED_4_ ≥ 95 Gy	52	30.8
BED_4_ < 95 Gy	117	69.2
**BEDGy_10_**		
BEDGy_10_ (median)	60.0 (36.0–87.2)	
BED_10_ ≥ 60 Gy	159	94.0
BED_10_ < 60 Gy	10	6.0
BED_10_ ≥ 65 Gy	72	42.6
BED_10_ < 65 Gy	97	57.4
**Chemotherapy**		
Yes	43	25.4
No	126	74.6
**CT Timing**		
Neoadjuvant	16	37.2
Adjuvant pre-RT	19	44.2
Concomitant	4	9.3
Adjuvant post-RT	4	9.3

BED = biological effective dose; CT = chemotherapy; FNCLCC = Fédération nationale des centres de lutte contre le cancer; IMRT = intensity-modulated radiation therapy; RT = radiotherapy; UPS = undifferentiated pleomorphic sarcoma; 2D-RT = two-dimensional-conformal radiotherapy; 3D-RT = three-dimensional-conformal radiotherapy.

**Table 2 cancers-16-01789-t002:** Univariate analysis.

	LC		DC		OS	
	5-Year	*p*-Value	5-Year	*p*-Value	5-Year	*p*-Value
**Sex**		0.999		0.739		0.774
Male	92.2%		75.6%		82.9%	
Female	92.2%		77.9%		84.6%	
**Age**		0.104		0.2		0.955
>60	89.0%		73.2%		83.9%	
<60	96.8%		82.0%		83.6%	
**Location**						
Upper arm	88.2%	0.287	73.9%	0.596	84.2%	0.919
Lower arm	94.0%		77.9%		83.5%	
Proximal	92.6%	0.761	79.8%	0.128	86.3%	0.128
Distal	90.9%		68.0%		76.3%	
**Depth**		0.756		0.072		**0.030**
Superficial	94.1%		95.2%		100%	
Deep	91.9%		73.9%		81.2%	
**Grade (FNCLCC)**		0.17		**<0.0001**		**0.005**
1	100%		95.4%		86.5%	
2	89.5%		85.1%		84.9%	
3	86.6%		54.6%		64.3%	
**Margin status**		0.633		0.146		0.557
R0	91.6%		70.3%		81.6%	
R1	92.3%		87.6%		86.0%	
R2	100%		100%		100%	
**Size**		0.453		0.113		0.413
≥10	94.9%		70.4%		86.6%	
<10	90.4%		81.3%		81.8%	
**TNM**		0.365		0.260		0.171
T1a	100%		92.3%		100%	
T1b	95.0%		83.8%		80.0%	
T2a	80.2%		83.3%		91.7%	
T2b	91.9%		72.8%		81.7%	
**Histological subtype**		0.618		**<0.0001**		**<0.0001**
Liposarcoma	98.0%		92.9%		96.4%	
UPS	88.8%		54.8%		70.8%	
Myxofibrosarcoma	92.2%		81.7%		85.8%	
Leiomyosarcoma	87.5%		79.8%		88.2%	
Synovial sarcoma	83.3%		66.7%		83.3%	
Undifferentiated sarcoma	72.9%		50.0%		77.8%	
Rhabdomyosarcoma	100%		0.0%		0.0%	
Other	92.9%		78.6%		62.9%	
**Liposarcoma subgroup**		**0.0007**		**<0.0001**		**0.0367**
Well-differentiated	100%		100%		100%	
Myxoid	100%		93.3%		93.3%	
Pleomorphic	100%		33.3%		66.7%	
Dedifferentiated	75.0%		83.3%		83.3%	
Unknown	100%		100%		100%	
**MDM2 amplification**		0.811		**0.005**		0.073
MDM2 (+)	94.0%		95.2%		92.9%	
MDM2 (−)	92.7%		68.1%		79.5%	
**Pre-operative RMI**		0.728		0.337		0.523
RMI (+)	92.1%		77.5%		84.7%	
RMI (−)	89.7%		69.1%		79.9%	
**RT Technique**		**0.006**		**0.013**		0.095
RT2D	100%		100%		100%	
RT3D	78.2%		63.7%		76.3%	
IMRT	95.7%		79.6%		85.6%	
**BED_4_**						
BED_4_ ≥ 75 Gy	92.6%	**0.015**	77.9%	**0.037**	83.3%	0.974
BED_4_ < 75 Gy	60.0%		40.0%		83.3%	
BED_4_ ≥ 85 Gy	88.8%	0.329	83.7%	0.082	84.6%	0.813
BED_4_ < 85 Gy	93.6%		71.6%		83.2%	
BED_4_ ≥ 95 Gy	90.0%	0.819	87.7%	**0.041**	88.3%	0.306
BED_4_ < 95 Gy	92.0%		71.9%		81.9%	
**BED_10_**						
BED_10_ ≥ 60 Gy	93.2%	**0.046**	78.3%	**0.022**	83.6%	0.771
BED_10_ < 60 Gy	71.4%		42.9%		87.5%	
BED_10_ ≥ 65 Gy	88.8%	0.222	71.6%	0.083	84.6%	0.80
BED_10_ < 65 Gy	94.8%		83.7%		83.1%	
**Chemotherapy**		0.234		0.537		0.647
CT (+)	87.3%		73.1%		86.0%	
CT (−)	94.2%		78.1%		83.0%	
**Relapse**						
Local relapse (+)	-	-	-	-	75.0%	0.313
Local relapse (−)	-	-	-	-	84.7%	
Distant relapse (+)	-	-	-	-	43.8%	**<0.0001**
Distant relapse (−)	-	-	-	-	96.1%	

Values in bold mean significance (*p* < 0.05). BED = biological effective dose; CT = chemotherapy; DC = distant control; FNCLCC = Fédération nationale des centres de lutte contre le cancer; IMRT = intensity-modulated radiation therapy; LC = local control; OS = overall survival; RT = radiotherapy; UPS = Undifferentiated pleomorphic sarcoma.

**Table 3 cancers-16-01789-t003:** Multivariate analysis.

**Distant Control**	**Hazard Ratio**	***p*-Value**
**Liposarcoma**	0.573 [0.188; 1.75]	0.328
**UPS**	1.76 [0.891; 3.48]	0.104
**Grade (FNCLCC)**	0.258 [0.0653; 1.02]	0.0529
**Depth**	0.14 [0.019; 1.02]	0.0529
**BED_10_ ≥ 60 Gy**	0.556 [0.212; 1.46]	0.233
**Overall Survival**	**Hazard Ratio**	***p*-Value**
**Liposarcoma**	0.372 [0.14; 0.988]	0.0473
**UPS**	1.06 [0.561; 1.99]	0.864
**Grade (FNCLCC)**	1.09 [0.406; 2.94]	0.86
**Depth**	0.148 [0.0201; 1.09]	0.0606
**Distant relapse**	3.94 [2.21; 7.01]	<0.0001

BED = biological effective dose; FNCLCC = Fédération nationale des centres de lutte contre le cancer; UPS = undifferentiated pleomorphic sarcoma.

**Table 4 cancers-16-01789-t004:** Location and proportion of metastases.

Metastasis Patients	*n* = 40	%
**Lung**	34	85
**Brain**	3	7.5
**Liver**	5	12.5
**Bone**	7	17.5
**Peritoneal carcinomatosis**	3	7.5
**Muscle**	4	10
**Lymph nodes**	6	15
**Other (adrenal, skin, spleen)**	3	7.5

## Data Availability

The data are available upon request due to privacy restrictions. The data presented in this study are available upon request from the corresponding author.

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
