# Peer review of "Factors Influencing Long-Term Local Recurrence, Distant Metastasis, and Survival in Patients with Soft Tissue Sarcoma of the Extremities Treated with Radiotherapy"

_cancers, 2024, doi:10.3390/cancers16101789_

Round 1
Reviewer 1 Report
Comments and Suggestions for Authors
This retrospective study entitled "Factors influencing long-term local recurrence, distant metastasis, and survival in patients with soft tissue sarcoma of the extremities treated with radiotherapy" performed by Lebas A et al. shows the efficacy of radiotherapy in combination with surgery for patients with non-metastatic extremity soft tissue sarcomas (ESTS), aiming to determine predictors influencing local and distant control, as well as overall survival.
Including 169 patients treated from 2007 to 2020, the study highlights the superior outcomes of intensity-modulated radiation therapy (IMRT) compared to three-dimensional conformal radiation therapy (3DRT) in achieving local control.
Liposarcoma, particularly well-differentiated subtypes, was identified as a favorable prognostic factor for overall survival, presenting lower rates of distant metastases and better survival outcomes.
Congratulations to the authors of the study, I just have some minor comments:
- In paragraph 2.4, please provide definitions of OS and AEs, or at least insert reference in line 108;
- In the Results section, lines 151,152, please detail chemotherapy data;
- Line 159, what kind of distant metastases did patients develop? Please detail;
- As part of surgical treatment for ESTS, please add this reference PMID: 34702592 to improve the quality of your manuscript.
Author Response
Dear Reviewer,
Thank you for your thoughtful review of our study on soft tissue sarcoma of the extremities treated with radiotherapy.
We are pleased that you found our study informative and valuable. Your feedback is greatly appreciated, and enhance the quality of our manuscript:
1) In paragraph 2.4, please provide definitions of OS and AEs, or at least insert reference in line 108;
Response : Definitions were added in the text.
OS was defined as the percentage of patients who are still alive at 5 / 10 years after diagnosis.
AEs were defined as post operative complications (wound complications such as hematoma, infection, necrosis) and lymphoedema; acute radiation-induced side effects (radiodermatitis and edema); chronic radiation-induced side effects (fibrosis, telangiectasias and fractures).
2) In the Results section, lines 151,152, please detail chemotherapy data;
Response : detailed data had been added.
Chemotherapy was administered to 43 patients (25.4%), mainly based on an-thracycline and. Of these, 16 patients (37.2%) received neoadjuvant CT, 19 patients (44.2%) received adjuvant CT prior to RT, 4 patients (9.3%) received concomitant CT, and 4 pa-tients (9.3%) received adjuvant CT after RT.
3) Line 159, what kind of distant metastases did patients develop? Please detail;
Response : Details of number and origin of metastases are mentioned line 219.
"Among the 58 metastases, most were located in the lung (82.5%), followed by the liver, bones, and lymph nodes (Table 4). "
4) As part of surgical treatment for ESTS, please add this reference PMID: 34702592 to improve the quality of your manuscript.
Response : The reference PMID: 34702592 was added ine the discussion.
Once again, we sincerely appreciate your constructive feedback and your time spent reviewing our work.
Best regards,
Arthur LEBAS
Reviewer 2 Report
Comments and Suggestions for Authors
Wide resection plus postoperative radiotherapy play an important role to control the local recurrence and distant metastasis, or even the overall survival. Questions remain:
1、Few patients underwent preoperative radiotherapy, what's your decision making to apply pre-or post-operative radiotherapy?
2、Patients who underwent incomplete surgery had less frequent metastases than did those with negative margins (p = 0.014). please explain.
3、Numerous previous published studies demonstrating that margin status is a crucial prognostic factor for local control, however, the authors mentioned that the current study did not replicate this finding. higher doses appeared to mitigate the adverse impact of positive margins. we think it is a controversial conclusion, due to this study limitations, it's better to revise to "higher dose may mitigate the adverse impact of positive margins"
Author Response
Dear Reviewer,
Thank you for your insightful comments on our study on soft tissue sarcoma of the extremities treated with radiotherapy.
1、Few patients underwent preoperative radiotherapy, what's your decision making to apply pre-or post-operative radiotherapy?
Response: This was due to old service habits as the patients included had been treated between 2007 and 2020, with surgeons preferring post-operative radiotherapy. Currently, practices tend to evolve with more pre-operative radiotherapy.
2、Patients who underwent incomplete surgery had less frequent metastases than did those with negative margins (p = 0.014). please explain.
Response : The observation that patients who underwent incomplete surgery had less frequent metastases compared to those with negative margins can be elucidated by considering the following factors:
Among patients with R0 resection and a higher incidence of metastases:
14% had grade 1 tumors
20.8% had liposarcomas, of which only 22.7% were well-differentiated
There was a higher prevalence of undifferentiated pleomorphic sarcomas (UPS) at 21.7%
In contrast, patients with R1/R2 resection and a lower incidence of metastases:
52.7% had grade 1 tumors
55% had liposarcomas, with 69.9% being well-differentiated
There was a lower prevalence of undifferentiated pleomorphic sarcomas (UPS) at 10%.
3、Numerous previous published studies demonstrating that margin status is a crucial prognostic factor for local control, however, the authors mentioned that the current study did not replicate this finding. higher doses appeared to mitigate the adverse impact of positive margins. we think it is a controversial conclusion, due to this study limitations, it's better to revise to "higher dose may mitigate the adverse impact of positive margins"
Response : Thank you for your comment with which we agree, the sentence is modified as you wrote it.
Once again, we sincerely appreciate your constructive feedback and your time spent reviewing our work.
Best regards,
Arthur LEBAS